An evolutionary Bi-LSTM-DQN framework for enhanced recognition and classification in rural information management

Deng Taiping 1
He Xi 1
Li Jiao 1
Ye Feifei 1
Tang Jingyang 2 tangjingyang_1573@163.com
1 School of Economics and Management, Hunan Applied Technology University , Changde, Hunan , China
2 School of Management, Changde College , Changde, Hunan , China
Asif Muhammad
Electronic publication date: 2025 Jun 27
Publication date: 2025
Volume: 11
Electronic Location ID: e2967
Received 2025 Feb 18; Accepted 2025 May 28
Copyright: © 2025 Deng et al.
Copyright year: 2025
Copyright holder: Deng et al.
License: This is an open access article distributed under the terms of the Creative Commons Attribution License, which permits unrestricted use, distribution, reproduction and adaptation in any medium and for any purpose provided that it is properly attributed. For attribution, the original author(s), title, publication source (PeerJ Computer Science) and either DOI or URL of the article must be cited.
License URL: https://creativecommons.org/licenses/by/4.0/

Keywords: Evolution algorithm, Reinforcement learning, Rural management, DQN

Funding: Subject of the 14th Five-Year Plan of Hunan Province “Research on the Cultivation Mode of ‘Huxiang Artisans’ in application-oriented colleges and universities in Hunan under the Background of National Rejuvenation through Science and Education”: ND248360 Key Research Project of Hunan Provincial Department of Education “Research on the Development Path of Yuan Li Digital Opera Culture Industry Based on the City”: 24A0767 General Education Project of Teaching Reform Research in Ordinary Colleges and Universities in Hunan Province “Making Labor a Mentally Healthy Lifestyle: Constructing a Model of Linking Labor with Culture and Education and Combining Education with Labor”: HNJG-2020-1269 Key Project of the Social Science Evaluation Committee of Changde City “Research on the Development of Changde’s “Three Rural” Issues from the Perspective of Rural Revitalization”: CSP22ZZ05 This work is funded by “Subject of the 14th Five-Year Plan of Hunan Province, Research on the Cultivation Mode of “Huxiang Artisans” in application-oriented colleges and universities in Hunan under the Background of National Rejuvenation through Science and Education”, the project number is ND248360; and supported by “Key research project of Hunan Provincial Department of Education, Research on the Development path of Yuan Li Digital Opera Culture Industry based on the city”, the project number is 24A0767; and supported by “General Education Project of Teaching Reform Research in ordinary Colleges and Universities in Hunan Province, Making Labor a Mentally healthy Lifestyle: Constructing a model of Linking Labor with Culture and Education and Combining education with Labor”, the project number is HNJG-2020-1269; and supported by “Key project of Social Science Evaluation Committee of Changde City, research on the development of Changde’s “three rural” issues from the perspective of rural revitalization”, the project number is CSP22ZZ05. The funders had no role in study design, data collection and analysis, decision to publish, or preparation of the manuscript.

==============================
As deep learning and reinforcement learning technologies advance, intelligent rural information management is transforming substantially. This article presents an innovative framework, the evolutionary bidirectional long short-term memory deep Q-network (EBLM-DQN), which integrates evolutionary algorithms, reinforcement learning, and bidirectional long short-term memory (Bi-LSTM) networks to significantly improve the accuracy and efficiency of rural information management, particularly for recognizing and classifying information relevant to farmers. The proposed framework begins with data preprocessing using disambiguation techniques and data complementation, followed by temporal feature extraction via a Bi-LSTM layer. It then employs a deep Q-network (DQN) to adjust and optimize weights dynamically. After feature extraction and weight optimization, evolutionary algorithms are used to select the optimal weights, enabling precise recognition and classification of conditions encountered by farmers seeking assistance. Experimental results indicate that the EBLM-DQN framework outperforms existing frameworks on public datasets and real-world applications, providing higher classification accuracy. This framework offers valuable technical support and a reference for future optimization and development of rural information management systems.

Introduction

Rural information management plays a crucial role in the country’s economic and social development, not only related to the vital interests of farmers and the prosperity of the rural economy but also directly affecting the country’s food security, ecological protection, and social stability. The importance of comprehensive rural information management is increasingly prominent, especially in achieving the strategic goals of common prosperity and promoting coordinated urban-rural development. With the continuous promotion of informatization and digital rural strategy, the traditional management method relying on manual recording and experience decision-making is gradually becoming limited and unable to meet the current development needs of refined and dynamic rural governance. Therefore, rural comprehensive information management has emerged and plays a key role in agricultural production, resource allocation, population structure monitoring, infrastructure evaluation, and other aspects. Rural comprehensive information management is shifting from static data processing to intelligent and real-time information fusion platforms, utilizing emerging technologies such as big data and artificial intelligence to efficiently collect, integrate, and analyze massive rural information. In this development context, farmer assistance has become a key and challenging issue in rural information management. Scientific and accurate identification of families needing help is a prerequisite for implementing various agricultural policies and achieving long-term poverty alleviation mechanisms. Traditional manual evaluation and empirical judgment methods are subjective and inefficient, making supporting large-scale, multi-dimensional data analysis tasks difficult. More and more research has begun to introduce machine learning and deep learning methods to improve recognition efficiency and scientificity in recent years. Automated classification and prediction can be achieved by modeling characteristic data such as farmers’ income, cultivated land area, education level, labor ability, and family structure. This type of method improves the accuracy and efficiency of recognition and provides data-driven decision support for policy formulation and resource allocation. However, in the face of the non-stationarity and environmental change characteristics of farmer data, existing static models still have problems such as slow response and limited generalization ability, and it is urgent to introduce more dynamic and flexible technical paths to solve them.

Due to the highly diverse and dynamically changing characteristics of farmer information, traditional machine learning and deep learning methods often face problems such as insufficient generalization ability and poor adaptability in practical recognition and assistance tasks (Graskemper, Yu & Feil, 2021). To address this challenge, this article introduces a reinforcement learning mechanism. It proposes an intelligent recognition method with adaptive optimization capability by combining it with a recurrent neural network (RNN) structure. Reinforcement learning can continuously learn and respond to changes by dynamically interacting with the environment, constantly adjusting strategies, and updating model weights. Compared to static modeling methods, the method proposed in this article can perceive changes in farmers’ information in real time, making it more flexible and accurate in identifying potential assistance targets. The innovation of this article lies in the first integration of reinforcement learning and deep temporal modeling, constructing an evolutionary bidirectional long short-term memory deep Q-network (EBLM-DQN) model suitable for rural dynamic management scenarios, realizing an intelligent and dynamic assistance decision-making mechanism, and providing a more forward-looking technological path for rural information management. The delineated contributions are as follows:

1. Addressing the exigencies of farmer assistance identification within the realm of information management, enhance the extant bidirectional long short-term memory (Bi-LSTM) methodology by integrating an evolutionary algorithm, culminating in the inception of evolutionary bidirectional long short-term memory (E-BiLSTM). Subsequently, fortify the model’s dynamic interactive capabilities via reinforcement learning utilizing a deep Q-network (DQN) network, thereby engendering the establishment of the EBLM-DQN network.

2. Employing the devised framework to achieve meticulous risk identification on financial risk analysis and prediction datasets, ascertain the optimal model, and facilitate seamless model migration during practical implementation.

3. This work introduces an EBiLSTM-based mechanism to identify farmers’ real-time assistance needs, addressing the gap in dynamic support recognition. By leveraging pre-training and refinement on actual management data, the model achieves over 95% accuracy, demonstrating its effectiveness.

The rest of this article is structured as follows: “Related Works” delves into the pertinent literature concerning the evolutionary algorithm and reinforcement learning. “Methodology” delineates the establishment of the proposed framework. “Experiments and Analysis” elucidates the experiment’s particulars and outcomes. Finally, the “Conclusion” is drawn in the concluding section.

Related works

Evolutionary algorithms and evolutionary neural network research

Evolutionary algorithms are a set of optimization algorithms inspired by natural selection and genetic processes inherent in biological evolution, widely utilized to approximate solutions for intricate problems. These algorithms leverage “individuals” within a “population” to represent potential solutions, iteratively refining these solutions. Classic examples include genetic algorithms (GA), genetic programming (GP), ant colony optimization (ACO), and more (Li et al., 2023). Evolutionary neural network algorithms enhance network performance by building on evolutionary algorithms. They begin with an initial population of neural network weights and evolve them towards better solutions through mutation, crossover, and selection. Crossover mutation creates new individuals with the potential for improved characteristics, while selection retains the best individuals and eliminates the weakest, ultimately leading to optimal weights for training the network.

Depending on the application, evolutionary neural network algorithms can be categorized into image classification, reinforcement learning, transfer learning, and time series prediction. In image classification, AmoebaNet-A is an algorithm that integrates tournament-selective evolutionary algorithms, bypassing manual design to enhance classification accuracy (Real, Aggarwal & Huang, 2019; Wu et al., 2023) significantly. Additionally, using differential evolution to create single-pixel adversarial perturbations demonstrates a novel approach to generating adversarial attacks by altering a single pixel, which can dramatically impact a network’s performance (Su, Vargas & Sakurai, 2019). For reinforcement learning, evolutionary neural network algorithms prevent local optima traps through an evolutionary approach, improving performance over traditional gradient descent methods. While gradient descent can sometimes mislead the agent towards suboptimal solutions, evolutionary reinforcement learning adopts non-gradient methods (like genetic algorithms) to find global optima, thus resolving issues that plague deep reinforcement learning (Such, Madhavan & Conti, 2018). Evolutionary neural networks for time series prediction often apply evolutionary algorithms to optimize the weights of LSTM models and their variants, yielding significant improvements in predictive performance. One approach embeds evolutionary algorithms directly into LSTM networks, leveraging their global optimization capabilities to capture trends in data like stock predictions (Zhang et al., 2025). An alternative embeds evolutionary algorithms into enhanced models for improved prediction, such as incorporating a competitive stochastic search algorithm into an LSTM with attention mechanisms to predict multivariate time series (Mnih et al., 2013). This allows evolutionary attention learning to be incorporated into LSTM models, enabling a model to realize competitive predictive capabilities by mining temporal relationships using shared weights.

Enhanced learning studies

Reinforcement learning (RL) is a key machine learning approach. An agent learns to map environmental states to behavioral actions to maximize rewards through interactions, effectively utilizing a reward-punishment paradigm. The agent strives to execute actions that yield higher rewards while avoiding those with lower rewards, embodying a highly adaptive form of autonomous learning. Mnih et al. (2015) introduced the DQN, merging DL with RL, marking the rapid development of deep reinforcement learning (DRL) (Xu et al., 2025; Gong et al., 2024). DRL offers an end-to-end perception and control system that combines the high-dimensional state space with the robust nonlinear approximation capabilities of DL neural networks. This system analyzes environmental data, extracts features, and perceives changes abstractly, overcoming RL’s limitations. It identifies optimal actions to maximize cumulative returns by exploring and refining learning strategies (Mnih et al., 2015; Wang et al., 2018; Ning et al., 2025). DRL algorithms fall into two categories: value function algorithms and policy gradient algorithms, based on their iterative methods. The value function algorithm iterates towards convergence, indirectly leading to the agent’s optimal strategy. Examples include the DQN, Dueling Deep Q Network (DDQN), and their variations (Wang, Li & Chen, 2023).

In contrast, policy gradient algorithms directly refine the agent’s optimal strategy by evaluating and improving policies and adjusting policy network parameters along the gradient. Examples include the Deep Deterministic Policy Gradient (DDPG), Twin Delayed Deep Deterministic Policy Gradient (TD3), Actor-Critic (AC), and Soft Actor-Critic (SAC) algorithms (Haarnoja et al., 2018). Li and colleagues modeled news article recommendations as a contextual bandit, enhancing click-through rates (Wei et al., 2024; Zhao et al., 2024). Chen et al. (2019), Huang et al. (2024) devised a reinforcement learning-based Top-K recommendation approach, improving long-term user satisfaction. Zheng et al. (2018), Chen & Pan (2019), Li, Li & Luo (2021) combined deep learning with reinforcement learning to develop a deep reinforcement learning recommendation system, demonstrating superior performance.

The research above demonstrates that evolutionary algorithms significantly enhance the optimization of neural network parameters, expediting model iterations and improving accuracy. Combining deep networks with reinforcement learning Q-networks enhances dynamic interaction with external environments, yielding superior real-time performance and interactivity when external data fluctuates. This is particularly advantageous for rural information management, where reinforcement learning methods facilitate efficient information management and provide robust critical data analysis.

Methodology

E-BiLSTM

Bi-LSTM is an advanced RNN that combines forward and backward LSTM units to better capture bidirectional dependencies in sequential data. Bi-LSTM is frequently employed in natural language processing, time series analysis, and tasks necessitating bidirectional contextual understanding. In rural information management, decision-making often hinges on historical data, such as determining whether villagers require assistance based on their living conditions. Thus, time series methods like Bi-LSTM are particularly suitable (Shi et al., 2022). Bi-LSTM consists of two independent LSTM layers that process data forward and reverse directions. This architecture enables comprehensive contextual information at any point in the sequence. Each LSTM layer contains multiple LSTM units, comprising input gates, forget gates, output gates, and memory units that regulate information flow. This effectively addresses the long-term dependency issues inherent in standard RNNs (Hamayel & Owda, 2021).

For the Bi-LSTM network, which has better applicability to the overall data compared to the traditional LSTM, the forward LSTM process is shown in Eq. (1):

(1) h→t=LSTM(xt,h→t−1),

where h→t is the positive hidden state at step t step of the forward hidden state, which depends on the current input xt and the previous hidden state h→t−1. The corresponding backpropagation process is then shown in Eq. (2):

(2) h←t=LSTM(xt,h←t−1),

where h←t is the reverse hidden state corresponding to the forward hidden state in Eq. (1). After completing the confirmation of forward and reverse hidden states, the connection of bi-directional LSTM hidden states can be obtained as.

(3) ht=[h→t;h←t],

where [;] denotes the connection operation. In this way ht contextual information at both ends of the input sequence is included simultaneously. To account for the variability and uncertainty inherent in rural data and enhance model performance, we propose a variation of the evolutionary algorithm that evaluates and selects an E-Bi-LSTM method for Bi-LSTM. The variation process involves setting different loss functions, allowing optimal loss values for the same batch of input data across different loss functions in the same model. By selecting several common loss functions, the training process can perform gradient descent based on the function yielding the best current loss value, thus enabling the model to utilize the optimal weights for the next evolutionary round.

The three chosen loss functions are the squared loss function, the absolute value loss function, and the Huber loss function (Meyer, 2021). Their mathematical definitions are given in Eqs. (4)–(6):

(4) Lsq(y,y^)=(y−y^)2.

The squared loss function measures the predicted y^ and the true value y squared error between the predicted value and the true value. It is more sensitive to large errors because the error is squared.

(5) Labs(y,y^)=|y−y^|.

The absolute value loss function measures the predicted y^ and the true value y absolute error between the predicted and actual values. It is insensitive to significant errors and performs better for minor errors.

(6) LHuber(y,y^)={12(y−y^)2if|y−y^|≤δδ|y−y^|−12δ2if|y−y^|>δ.

The Huber loss function combines the advantages of squared Loss and absolute value loss. For smaller errors (less than δ), it is the same as squared Loss, while for larger errors (greater than δ), it is similar to absolute value loss, which reduces sensitivity to outliers. The reason for choosing these three loss functions is to balance stability and robustness during the model training process. The squared loss function (MSE) is susceptible to situations where the prediction error is small, which helps to accelerate the convergence speed of the model, but it is susceptible to significant interference in the presence of outliers. The absolute value loss function (MAE) is more robust to outliers and can balance the impact of noise on model training, but may converge slowly during gradient updates. The Huber loss function combines the advantages of the first two, using a square term when the error is small and a linear term when the error is large, balancing stability and anti-interference ability. Therefore, this article introduces three different loss functions with varying characteristics for dynamic evaluation and selection, which can improve the adaptability and overall performance of the model in different data states.

During the evaluation process, to streamline model performance, we use a function that directly sorts the loss values to evaluate the model. By fixing the mode of gradient descent and selecting the smallest loss value across different loss functions, we can identify the current optimal weight optimization function. The minimum loss function, Loss, can be represented as:

(7) Loss=min(Lsq(y,y^),Labs(y,y^),Lhuber(y,y^)).

Based on this, the optimal loss function corresponding to the best variation strategy is selected to derive the optimal population, facilitating the most effective analysis and adjustment of network parameters. The overall process, which includes the selection of the optimal loss function, variation strategy, and network parameter adjustment, is illustrated in Fig. 1.

Figure 1 The framework for the E-BiLSTM.

DQN

DQN is a deep reinforcement learning algorithm that merges deep learning with Q-learning, employing neural networks to approximate Q-functions. It tackles the obstacle of applying conventional Q-learning in high-dimensional state spaces. DQN’s core idea is to use neural networks to approximate action-value functions (Q-functions) to improve policy learning in complex environments. In a typical DQN, past experiences (states, actions, rewards, and following states) are randomly sampled from an experience replay buffer to train the network (Li et al., 2022). This strategy reduces the correlation within the training data, improving training results. DQNs leverage the target network and the primary network, which calculate the target and estimated Q-values, respectively. The target network’s parameters are periodically copied from the leading network to ensure stable target values. A typical DQN network structure primarily involves updating Q-values through interaction and making predictions via the intelligent agent. Traditional Q-learning formulates Q-value updates as follows:

(8) Q(s,a)←Q(s,a)+α(r+γmaxa′Q(s′,a′)−Q(s,a)),

where s and s′ are the current state and the next state, respectively, and a is the current action, and r is the reward, α is the learning rate, and γ is the discount factor. In the practical application of this article, taking the observations of farmers updating information in the relevant departments in the rural information management system as an example, the interaction update phase is the phase of the intelligent agent to perform the recommendation task after obtaining the available reward prediction model, the intelligent agent interacts with the real and simulated environments at the same time, and learns and updates its recommendation strategy during the interaction, and the diagram of the interaction update phase is shown in Fig. 2.

Figure 2 The interaction and update process for the DQN model.

At each interaction time step, the interaction between the intelligence and the environment will generate interaction data (ob,A,R), where ob represents the current state of the environment. At the end of each round of interaction, the intelligent agent will use the interaction data to train and update the decision model and the state-value model by a one-step Actor-Critic method, in which the parameters of the decision model and the state-value model are trained using the SGD optimizer. In reinforcement learning, intelligent agents often trade between immediate and long-term rewards. A higher emphasis on immediate rewards tends to lead the agent towards a more conservative action strategy (Carta et al., 2021). Therefore, in this article, a dynamic constraint term is added to the loss function of the algorithm based on the model-picking approach, as shown in Eq. (9):

(9) θi+1←θi+α((Ri+γv^w(St+1)−v^w(St))∇θln⁡πθ(Si,Ai)−ef∑i|A|∇θ(πθ(Ai,St)lnπθ(Ai,Si)))⋯

where πθ(St,At) denotes the state. The recommendation result, influenced by this constraint term, is also recognized as an enhancement in the effectiveness of farmer assistance suggestions within rural information management. Throughout the process, the constraint term itself negatively correlates with the diversity of the agent’s recommendation outcomes. A more conservative recommendation strategy leads to less diversity in the recommendation results, as the agent’s learning objective focuses on minimizing. By adding the loss function of the dynamic constraint term, the agent’s recommendation strategy aims to optimize future rewards, thereby enhancing the diversity of recommendation outcomes.

EBLM-DQN

After outlining the E-BiLSTM network and introducing the corresponding reinforcement learning DQN network, we integrated the two to create a new EBLM-DQN tailored for rural information management. This network is designed to assess the economic status of rural residents based on their information and to recommend relevant aid policies. The overall framework is illustrated in Fig. 3.

Figure 3 The framework for the EBLM-DQN.

In Fig. 3, the data preprocessing work is first carried out, and the model data is prepared through word segmentation technology and data completion. On this basis, we further train the model and improve its performance by adjusting the number of hidden layers during the model setup process. After the output of the BiLSTM layer, the results are input into the DQN network to simulate the dynamic interaction process between the agent and the environment. In each iteration, the model selects the optimal strategy, which is the combination of feature weights based on the current state, and continuously adjusts the plan according to environmental feedback, including recognition accuracy, loss changes, etc., to achieve optimal action selection. Introducing reinforcement learning mechanisms in this process enhances the model’s adaptability and generalization ability. Subsequently, the E-BiLSTM method proposed in the first section was used for iterative weight updates and optimal selection. Finally, the function with the minimum Loss was selected to construct the overall model named EBLM-DQN.

Experiments and analysis

Dataset and data preprocessing

In rural information management, the core is a time-series decision-making problem, such as deciding whether villagers require assistance based on historical data. To better evaluate the model’s performance and analyze its capabilities, we tested it using public datasets relevant to the problem’s characteristics. Transfer learning techniques were applied to initialize and test the model on real-world data. This study used the widely used Credit Card Fraud Detection Dataset (https://zenodo.org/records/7395559; doi: 10.5281/zenodo.7395559) to assess the model’s performance.

This dataset includes 28 features after PCA dimensionality reduction (labeled V1 to V28) and two additional features: transaction amount and transaction time (Alarfaj et al., 2022). The target variable is a binary label indicating whether a transaction is fraudulent. With 284,000 transaction records, only 492 (approximately 0.17%) are fraudulent, making it a binary classification problem similar to assessing the need for assistance in rural information management. The dataset’s typical feature data from V1–V9 is illustrated in Fig. 4.

Figure 4 The feature for the public dataset.

In Fig. 4, the horizontal axis represents the sample number, which is the position of each transaction record in the dataset and is used to reflect the changes in features with the order of transactions. The vertical axis represents the numerical values of corresponding features. After PCA dimensionality reduction, these features do not have direct physical meaning but reflect the main change patterns in the original high-dimensional data. The figure shows that there may be significant differences in different features between normal and abnormal transactions, which provides valuable information for subsequent fraud recognition models. This analysis method can also be applied to the “whether to assist” discrimination problem in rural information management. Temporal feature extraction was performed to isolate time-dependent variables, using sliding windows to capture sequential patterns and trends. Dimensionality reduction was implemented using principal component analysis (PCA), retaining components with eigenvalues above a threshold to ensure computational efficiency. For numerical features, min-max normalization scaled values to 0 to 1, while categorical features were transformed using one-hot encoding. Text-based data underwent tokenization and stop-word removal, with token sequences mapped to numerical vectors using word embeddings.

The dataset was split into training, validation, and testing subsets at a ratio of 70:15:15, ensuring random but stratified distribution to maintain class balance. Cross-validation was performed on the training set to tune hyperparameters. Finally, the data was organized into batch sizes optimized for the Bi-LSTM architecture, and all preprocessing results were stored in standardized formats for reproducibility.

The experiment setup

The computing infrastructure for implementing the EBLM-DQN framework consisted of a high-performance hardware and software setup. The hardware included an Intel Xeon Platinum 8276 processor with 28 cores, an NVIDIA Tesla V100 GPU with 32 GB VRAM, 256 GB DDR4 RAM, and a 2 TB NVMe SSD for high-speed data processing and storage. The system operated on Ubuntu 20.04.6 LTS with Linux Kernel 5.15. Python 3.9 was the primary programming language, with TensorFlow 2.13.0 and PyTorch 2.0 employed for the deep learning components, including Bi-LSTM and reinforcement learning implementations, respectively. Data preprocessing and analysis were conducted using Scikit-learn 1.3, NumPy 1.25, and Pandas 2.0, while DEAP was leveraged for evolutionary algorithm optimization. The reinforcement learning component relied on Stable-Baselines3, and GPU acceleration was achieved using CUDA 11.8. Results were visualized with Matplotlib 3.8 and Seaborn 0.13, and task scheduling was managed through the Slurm Workload Manager in a clustered computing environment.

We trained the model with the overall process described in Algorithm 1. For evaluation, given that it is a binary classification task, we used the accuracy index to assess the model’s performance.

Algorithm 1 EBLM-DQN.

Input:	
  Credit Card Fraud Detection Dataset	
  Initialization.	
  Define the BiLSTM-DQN and Loss variation	
  Define initial weights, optimizer, batch size and max epochs.	
  Model training.	
  for i=1 to max epoch	
  data fed to BiLSTM-DQN.	
  Lsq, Labs and Lhuber calculation	
  switch min Loss	
  end switch	
  Parameters Fine tuning	
  Weights update ← min loss	
  Compute accuracy	
  Save the optimal model	
  End for	
Output:	
  Trained EBLM-DQN	

Assessment metrics

The assessment metrics used in this study include classification accuracy, loss evaluation, and confusion matrix analysis. Classification accuracy was chosen as a primary metric to measure the model’s capability to correctly identify both high-risk and no-risk instances in the datasets. Loss evaluation, including squared Loss, absolute value loss, and Huber loss, was employed to optimize the training process dynamically and ensure robustness in handling varying error magnitudes. The confusion matrix was used to evaluate the model’s detailed performance across different classes, providing insights into its precision and recall. These metrics were justified based on their effectiveness in assessing binary classification tasks and their ability to capture nuanced performance differences in both public and practical datasets.

Model comparison and ablation experiment

After completing the model training and data processing, we tested the performance of the proposed model, explicitly focusing on the evolutionary algorithm’s effectiveness, E-BiLSTM. To analyze the impact of loss selection, we compared models with and without loss selection, observing changes in Loss during training, which are illustrated in Fig. 5.

Figure 5 The training process for the model with evolution and without evolution.

In Fig. 5, we observe that after loss selection, the overall loss value is consistently below the average and remains lower during subsequent stable fluctuations. Concurrently, examining the changes in accuracy, the final accuracy is also improved, attributed to iterative loss optimization. This indicates that the evolutionary loss function can more effectively guide model learning during the training process, helping to reduce the risk of overfitting and improve generalization ability. Compared with the average Loss, evolutionary Loss exhibits faster convergence speed and more stable optimization trend in multiple stages, verifying its superiority in tasks with complex features and imbalanced samples. Ablation experiments were conducted for different model states, with results presented in Fig. 6.

Figure 6 The ablation experiment for the risk recognition result.

In Fig. 6, we conducted ablation experiments on the proposed EBLM-DQN model, sequentially removing the reinforcement learning module (No DQN) and weight evolution mechanism (No evolution). The complete model, DQN removed, and evolution removed confusion matrix results are shown from left to right. The results showed that EBLM-DQN achieved a recognition accuracy of 0.947 in the “High risk” category, significantly better than 0.921 after removing DQN and 0.901 after removing evolution, demonstrating stronger anomaly detection capability. At the same time, in recognition of the “No risk” category, EBLM-DQN also reached 0.963, which is better than the other two schemes (neither exceeding 0.94). This indicates that the DQN strategy and evolutionary mechanism are key in improving model performance, especially in identifying high-risk categories under imbalanced samples.

After the ablation experiments, we further examined the number of hidden layers in the Bi-LSTM used. Each hidden layer consisted of 64 cellular units, and the model’s performance was tested with different numbers of layers. The results are presented in Table 1 and Fig. 7.

Table 1 The average accuracy with different BiLSTM layers.

Layer	Proposed	No DQN	No evolution	
1	0.912	0.905	0.891	
2	0.923	0.911	0.899	
3	0.955	0.929	0.907	
4	0.951	0.928	0.905	
5	0.953	0.913	0.911	

Figure 7 The result for the average accuracy.

In the model testing process outlined in Table 1, it was found that setting the number of hidden layers to three yields higher overall recognition accuracy, regardless of model improvement. Additionally, increasing the hidden layers beyond three does not significantly improve accuracy. Therefore, a shallow hidden layer setup is effective for the dataset used, which has low data dimensionality.

Based on this, a comparison and analysis of the recognition accuracy of different methods on this dataset are presented in Fig. 8.

Figure 8 The method comparison result on credit card fraud detection dataset.

Figure 8 shows that the proposed model achieves superior recognition accuracy, owing significantly to foundational methods like Bi-LSTM in identifying high-risk and no-risk instances. This model outperforms the second-best approach by nearly 5%, indicating that incorporating reinforcement learning and iterative weight evolution notably enhances the method’s performance. Consequently, this article utilizes the optimal model from the public dataset for practical testing and problem analysis, with the specific process to be elaborated in the following subsection.

Model application in rural information management

After evaluating several candidate architectures, including traditional LSTM, BiLSTM, and EBiLSTM variants, the EBiLSTM model was selected based on its superior performance in capturing temporal dependencies and achieving higher validation accuracy. Transfer learning was employed to initialize the network weights using a pre-trained model trained on a related rural management dataset. This approach allowed the model to retain generalized temporal knowledge and adapt more efficiently to the specific task of identifying farm household assistance needs within the integrated rural information management process. To understand the characteristics of regional farm households, we collected information from approximately 190 households and derived their main features using the system weights. The five primary features of the extracted data are shown in Fig. 9. The features in Fig. 9 include income situation, household population structure, cultivated land area, education level, and labor force status. The figure shows that each feature fluctuates significantly between different samples, reflecting the heterogeneity characteristics among rural households. These features, after standardization, are used for subsequent modeling and object recognition tasks, with good discriminative ability and practical representativeness.

Figure 9 The practical application feature representation.

In this article, we collected data from farmers encompassing 20 different dimensions. Using this data, we addressed the issue of identifying which farmers require assistance, utilizing various methods. The accuracy of the assistance recognition in the practical application is presented in Table 2 and Fig. 10.

Table 2 The accuracy for the assistance recognition in the practical application.

Method	Proposed	Bi-LSTM	LSTM	RNN	Random forest	SVM	
Assistance	0.967	0.954	0.938	0.915	0.889	0.892	
No assistance	0.976	0.965	0.947	0.939	0.941	0.902	

Figure 10 The accuracy for the assistance recognition in the practical application.

The results reveal that the method proposed achieves a recognition accuracy exceeding 0.96 for determining whether farmers need assistance. Due to the smaller dataset and simpler dimensions, the overall advantage is not as pronounced as the public dataset but still surpasses traditional methods. This method has high recognition accuracy for “no-assistance” users, mainly due to the distribution characteristics of data and the model’s effective fitting of mainstream features. In rural datasets, “no-assistance” samples usually occupy the majority and have more obvious and concentrated characteristic patterns, such as stable income, better education, or lower debt, making it easier for the model to learn its discriminative boundaries. Meanwhile, the method proposed in this article enhances the model’s ability to focus on the main features by introducing reinforcement learning and feature evolution mechanisms, resulting in more robust performance in most class recognition. Therefore, even with a small sample size, high-precision identification of “no-assistance” farmers can still be achieved. Based on this, we conducted model ablation experiments on our real dataset, testing the model’s performance without reinforcement learning and evolutionary algorithms. The optimal model was validated using ten-fold cross-validation, and the results are shown in Fig. 11.

Figure 11 The ablation experiment for the assistance recognition.

In Fig. 11, it is evident that the model proposed exhibits better stability in recognition accuracy for both assisted and unassisted cases under identical testing conditions. The overall performance surpasses models that don’t utilize DQN and evolutionary algorithms. Therefore, the EBLM-DQN proposed provides superior decision support in the intelligent management of rural information, effectively recognizing whether farmers need assistance.

Discussion

To effectively model the rural system for policy learning, the framework incorporates a set of representative variables reflecting rural development’s multidimensional nature. These variables include agricultural output, rural infrastructure development, education attainment levels, healthcare availability, and employment rates. The EBiLSTM module captures the temporal dynamics and interdependencies among these indicators, enabling the framework to learn from historical patterns and forecast potential trends. At the base level, the DQN operates on the state space defined by these variables, where each state represents a specific configuration of rural development conditions. Policy actions, such as investment allocation or program implementation, are treated as actions in the DQN framework. The reward function is designed to align with long-term development goals, such as reducing inequality or improving quality of life, allowing the agent to iteratively learn policies that yield optimal and sustainable outcomes in complex rural environments. EBLM-DQN optimizes network prediction classification by combining evolutionary algorithms, reinforcement learning, and Bi-LSTM, offering clear advantages over traditional temporal networks like LSTM. First, EBLM-DQN merges evolutionary algorithms with reinforcement learning to dynamically adjust network weights, allowing the network to adapt more flexibly to data changes during training. Evolutionary algorithms excel in exploring optimal weight combinations for search-optimization problems, while reinforcement learning enhances EBLM-DQN’s strategic adaptability in dynamic data environments. Moreover, Bi-LSTM’s bidirectional nature enables simultaneous consideration of preceding and succeeding time-series relationships, improving the extraction of temporal features. EBLM-DQN’s blend of Bi-LSTM features with the benefits of evolutionary algorithms and reinforcement learning leads to substantial improvements in network predictive classification performance. As a result, the model achieves superior results in risk prediction tasks using public datasets and identifying farmers needing help using a self-built dataset.

In rural information management, EBLM-DQN significantly assists in various tasks. EBLM-DQN can classify and predict rural information data using its robust feature extraction and dynamic weight adjustment capabilities in data preprocessing and model training. In rural poverty alleviation and assistance targeting, EBLM-DQN effectively identifies key targets from large data volumes, enabling precise and practical assistance. For future applications, it’s essential to consider the model’s complexity and the data required for training. Due to the high computational complexity of evolutionary algorithms and reinforcement learning, the model’s training process might be time-consuming, necessitating thorough preprocessing and sample selection to reduce training time. Model parameters must also be finely tuned for specific applications to leverage their performance benefits fully. By further optimizing evolutionary algorithms and reinforcement learning strategies, EBLM-DQN can maintain high adaptability and prediction accuracy in dynamic environments, offering strong support for rural information management.

The proposed EBLM-DQN framework, while demonstrating significant improvements in classification accuracy and adaptability, presents some limitations specific to its structural design. Integrating Bi-LSTM and DQN increases the model’s complexity, making it computationally intensive and potentially challenging to deploy on systems with limited resources. Secondly, the reliance on Bi-LSTM for temporal feature extraction may encounter performance bottlenecks when processing highly complex or non-sequential data patterns, as it is inherently tailored for time-series data. Additionally, although practical, evolutionary algorithms for weight optimization can lead to convergence issues when applied to high-dimensional data spaces, potentially affecting stability during training. Finally, while enhancing robustness, the framework’s dynamic loss function selection might introduce overfitting risks if not carefully calibrated for diverse datasets. Addressing these structural challenges could involve exploring lightweight architectures, hybrid optimization strategies, and advanced regularization techniques to enhance scalability and generalization. Compared to conventional methods without the integrated Bi-LSTM and DQN architecture, the EBLM-DQN framework offers significantly enhanced adaptability and classification performance, particularly in dynamic environments. However, traditional approaches benefit from lower computational overhead and simpler training pipelines, making them more suitable for resource-constrained or real-time applications.

Conclusion

In this article, we introduce an EBLM-DQN framework that merges evolutionary algorithms and DQNs to address the problem of recognizing and classifying rural farmers in rural information management, aiming to achieve higher accuracy in managing farmer assistance and information. The framework integrates the bidirectional feature extraction of Bi-LSTM, multi-level reinforcement learning strategies, and the optimization capabilities of evolutionary algorithms. Public datasets and practical application tests indicate that the framework’s classification accuracy and efficiency surpass traditional machine learning and deep learning methods. The public credit risk identification dataset achieves an average recognition rate of 95.5% for identifying the presence or absence of credit financial risk. The model’s recognition accuracy on the local farmers’ dataset exceeds 96%, enabling practical assessment of farmers’ economic status to determine whether targeted assistance is necessary.

Future research will further enhance the model’s dynamic interaction capabilities and improve farmer identification and classification by incorporating multi-dimensional data, such as geographic and climate information. Additionally, establishing a standardized rural information management research dataset will provide valuable references for researchers and managers, supporting rural information management’s optimization and intelligent development.

Supplemental Information

Supplemental Information 1 This is the code.

Additional Information and Declarations

Competing Interests

The authors declare that they have no competing interests.

Author Contributions

Taiping Deng conceived and designed the experiments, performed the computation work, prepared figures and/or tables, and approved the final draft.

Xi He conceived and designed the experiments, prepared figures and/or tables, and approved the final draft.

Jiao Li conceived and designed the experiments, analyzed the data, authored or reviewed drafts of the article, and approved the final draft.

Feifei Ye performed the experiments, analyzed the data, authored or reviewed drafts of the article, and approved the final draft.

Jingyang Tang performed the experiments, performed the computation work, prepared figures and/or tables, authored or reviewed drafts of the article, and approved the final draft.

Data Availability

The following information was supplied regarding data availability:

The code is available in the Supplemental File.

The data is available at Zenodo: Luqi Liu. (2022). Credit Card Fraud Detection [Data set]. Zenodo. https://doi.org/10.5281/zenodo.7395559.

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
