# Peer review of "An evolutionary Bi-LSTM-DQN framework for enhanced recognition and classification in rural information management"

_PeerJ Computer Science, doi:10.7717/peerj-cs.2967_

## Round 0.1 · original submission · Major Revisions

Dear authors,

Your paper has been reviewed by the experts in the field and you will see that they have a couple of good suggestions for improvements. please take a look carefully and revise the paper accordingly.

Please submit a detailed response and justification for each point. Also, improve the language quality.

After that, we will reconsider it for re-evaluation.
Thank you

Reviewer 1 ·

Basic reporting

The objective of presented research is rural governance by using the farmer’s/rural data for historical data based farmer’s assistance. The focus is on comprehension, management, analysis of the data for the said purpose. A framework is proposed with the claimed enhancements to improve the accuracy, efficiency and adaptability of rural information system. Bi-LSTM framework is used. Claimed enhancement comprises of adding/adapting out of three existing loss function and selecting the optimal one. For training purpose Deep Q Network is used as a justification of reinforcement learning application without detailed discussion of reinforcement problem modeling as well as environment model required in RL.

Experimental design

The setup for experimentation consisted of high end processing units including 28 core CPU and GPU with enough RAM capacities. Python (multiple packages) language is used to process and visualize the data. As per the presented research the focus is on time series based rural data but the experimentation is performed on bank transactions data, comprising of 284000 rows/records. Initially 28 features are reduced using PCA into 9 features, represented in the Figure 4. But what the Figure 4 actually explains is not explicitly described? There should be at least some text dedicated to the explanation of the Figure 4, what is on the both of the axes as well as the what is thinness & thickness of the graphs? etc.

Validity of the findings

As per the graphs in Figure 5 of loss as well as accuracy, drawn between evolution / average loss and accuracy, the evolutionary loss functions performed well. There should be some elaboration due to which the evolutionary loss function performed well.
In Figure 6, three confusion matrix are shown, each representing accuracy, false alarm, missing alarm, etc. The difference between the values in black boxes in very small whereas the difference in white boxes is comparatively large. The titles of these boxes must be explicitly mentioned or they must be explained in the text discussing the confusion matrixes. Moreover, again possible reasoning is required for the improvement of the result by using the proposed solution.
Three assessment metric are used for evaluation purposes, namely, classification accuracy, loss function evaluation and confusion matrix. The classification accuracy and confusion matrix discussion is not found the paper.
In Figure 9 the purpose of the farm house hold features is not clear. Elaboration is required with respect to the proposed solution.
In Figure 10, why No assistance achieves high accuracy of 0.98 as compared to the different models in which proposed model achieves the highest accuracy?

Additional comments

line 64 & 65: reference required
lines 68 & 69 need elaboration
178 does not make sense
185 typo
In the paper evolutionary algorithms are claimed and proposed loss functions out of which optimal one will be selected for doing the gradient descent. It is unclear where the evolutionary algorithm fits in as per claim
244 What is intelligent body?
245 The real & simulated environments requires more elaboration
249 not clear meaning, explicitly confirm if A is action and R is reward
256 It is claimed that (short & long term reward inclination is avoided using some dynamic selection as with the loss function). Along with this, detailed elaboration of equation 9 is required
270 No detailed problem modeling of rural management for EBLM-DQN is provided.
270 “This network is designed to assess the economic status of rural 272 residents based on their information and to recommend relevant aid policies.”
Unclear how aid policies will be inferred from the proposed RL based deep learning solution and also what is the difference or improvement if normal deep networking solution would have been used.
279 “which dynamically interacts with the environment and forms different loss function weight optimizations” Elaboration of the sentence is required.
315 Elaboration is also required for the effect of cross-validation on training set for hyper-parameters tuning.
Moreover the model of the rural system is not discussed that would have been used for policy learning etc.

Reviewer 2 ·

Basic reporting

Your paper presents an evolutionary algorithm framework based on Bi-LSTM and DQN, demonstrating innovative applications in rural information management. However, the paper is somewhat verbose, with repeated explanations of certain concepts, which affects overall readability. Below are some specific suggestions for improvement:
1. In the introduction (Section 1), a significant portion is devoted to discussing the importance of rural information management (paragraphs 1 and 2). It is recommended to shorten this part and focus more on the limitations of current methods and your innovations.
2. Expressions such as "Integrated rural information management stands as a linchpin in contemporary agriculture and rural advancement" are overly complex, making it difficult for readers to understand. A more straightforward phrasing such as "Modern rural information management is a key factor in agricultural and rural development." would be preferable.
3. In Section 3.1, equations (4)–(7) lack clear explanations of their derivation, making it difficult for readers to understand their role. It is suggested to add one or two sentences after each equation to explain its physical significance, such as why Huber loss is chosen and how it specifically contributes to your method.
4. In Section 3.3, while you introduce the EBLM-DQN framework, you do not explicitly explain how the information flow between Bi-LSTM and DQN interacts.
5. There is some content redundancy between Section 4.2 "Experiment setup" and Section 4.3 "Assessment Metrics", particularly regarding data preprocessing methods.
6. It is recommended to include a convergence curve for DQN (e.g., a reward change curve) in the experimental section to validate the model’s stability.
7. The paper inconsistently uses both "evolution algorithm" and "evolutionary algorithm." These should be standardized to "evolutionary algorithm" to align with academic conventions.
8. Similarly, "Rural management" sometimes appears as "rural information management." To avoid confusion, maintain consistency throughout the paper.
9. The paper does not evaluate the algorithm's runtime, which is crucial for real-world applications. It is recommended to add a comparison of training and inference times for different models to assess the feasibility of practical deployment.
10. Certain sections of the paper excessively use academic jargon, making it more difficult to read. For example,
1. "Integrated rural information management stands as a linchpin in contemporary agriculture and rural advancement."
2. Could be rewritten as:
3. "Integrated rural information management is essential for modern agriculture and rural development."

Experimental design

In Section 3.1, equations (4)–(7) lack clear explanations of their derivation, making it difficult for readers to understand their role. It is suggested to add one or two sentences after each equation to explain its physical significance, such as why Huber loss is chosen and how it specifically contributes to your method.
In Section 3.3, while you introduce the EBLM-DQN framework, you do not explicitly explain how the information flow between Bi-LSTM and DQN interacts.
There is some content redundancy between Section 4.2 "Experiment setup" and Section 4.3 "Assessment Metrics", particularly regarding data preprocessing methods.

Validity of the findings

The findings are valid

---

## Round 0.2 · Minor Revisions

Dear authors,

Thank you for your resubmission. after careful consideration and based on the input from experts, we are of the view that although the paper has been improved from the previous version. but still needs some more improvements as mentioned in the reviewer's comments. Please incorporate those suggestions and re-submit for next round. thank you

Reviewer 1 ·

Basic reporting

The authors have revised the paper as per the raised questions, successfully, almost 90%.
The raised question involved which are successfully revised, some them are given below:
1- Explanation/Elaboration of different figures and graphs.
2- Different Key Points, e.g., confusion matrix, assessment metric, required references, typos, etc.
3- Elaboration of different existing models, including, loss functions, RL, etc.
4- Etc.

Experimental design

No questions were raised on experimental Design, initially.

Validity of the findings

No questions were raised on Validity of the findings, initially.

Additional comments

Following are the previously raised questions that are not answered or not referred, if the concerned revision is incorporated?
- Unclear how aid policies will be inferred from the proposed RL based deep learning solution and also what is the difference or improvement if normal deep networking solution would have been used.
- Moreover the model of the rural system is not discussed that would have been used for policy learning etc. The required discussion involve which variables from the rural development are modelled in EBiLSTM and how DQN is applied (at the base level).
In the revised paper following are the questions:
Line # 143 - 147 : Contribution # 3: It is not clear if is a contribution or application of the presented research work. If it is application then many other applications can be discussed and described. Contribution must involve the updated solution or new mechanism for some unaddressed problem which can be EBiLSTM.
Line # 498-499, What is model selection here and in which role transfer learning is used here?

Reviewer 2 ·

Basic reporting

The manuscript is accepted for publication.

Experimental design

The manuscript is accepted for publication.

Validity of the findings

The manuscript is accepted for publication.

---

## Round 0.3 · accepted · Accept

Dear authors

Thanks for your resubmission and improvements made in 2nd round. We are pleased to inform you that your manuscript is judged scientifically sound and is being recommended for publication. Congratulations